# Effect of Culling Management Practices on the Seroprevalence of Johne’s Disease in Holstein Dairy Cattle in Central Italy

**DOI:** 10.3390/vetsci9040162

**Published:** 2022-03-28

**Authors:** Martina Crociati, Luca Grispoldi, Athanasios Chalias, Maurizio Monaci, Beniamino Cenci-Goga, Lakamy Sylla

**Affiliations:** 1Dipartimento di Medicina Veterinaria, Università degli Studi di Perugia, 06126 Perugia, Italy; martina.crociati@unipg.it (M.C.); a.chalias@gmail.com (A.C.); maurizio.monaci@unipg.it (M.M.); beniamino.cencigoga@unipg.it (B.C.-G.); lakamy.sylla@unipg.it (L.S.); 2Centre for Perinatal and Reproductive Medicine, University of Perugia, 06126 Perugia, Italy; 3European Food Safety Authority, EU-FORA Programme, 43126 Parma, Italy; 4Department of Paraclinical Sciences, Faculty of Veterinary Science, University of Pretoria, Onderstepoort 0110, South Africa

**Keywords:** paratuberculosis, Johne’s disease, seroprevalence, culling strategy, productive life

## Abstract

A study was performed in Umbria, central Italy, to find out whether different culling strategies adopted by farms to control Johne’s disease (JD) infection exerted effects on the seroprevalence in dairy cattle. Fifty Fresian dairy herds in the Perugia and Assisi districts were visited and an audit of herd management was conducted. Among the 50 herds, 20 were selected for the consistency of management practices and, according to the culling strategy, two groups were created: group A (aggressive culling protocol, with average herd productive life <1100 days) and group B (lower culling rate, with productive life greater than 1500 days). The presence of antibodies to *Mycobacterium avium* subspecies *paratuberculosis* (*Map*) in the serum was determined using a commercial enzyme-linked immunosorbent assay (ELISA) kit. It was found that 3.3% (*n* = 14) of the cows of group B (*n* = 422, from 17 herds) were positive for *Map* antibodies, in comparison with 5.7% (*n* = 21) of the cows from group A (*n* = 366, from three herds). The odds ratio from multiple logistic regression (adjusted odds ratio 2.446, 95% confidence interval 0.412 to 14.525) showed that Johne’s disease prevalence in herds with a greater productive life was not higher than in herds with typical modern management characterized by more aggressive culling. This is a significant finding, indicating that aggressive culling may not be necessary. Current JD control recommendations are derived from data obtained in high-prevalence paratuberculosis areas (northern Europe, including northern Italy), while methods of information transfer to dairy farms in low-prevalence areas should be reassessed to ensure that the correct measures, including basic calving management and calf-rearing practices, are thoroughly implemented. Using the manufacturer’s suggested cut-off for a positive ELISA test and the sensitivity and specificity claimed, the overall true prevalence in Umbria dairy cattle was calculated as 7% (95% confidence interval 5.2% to 8.8%).

## 1. Introduction

Johne’s disease is a chronic granulomatous enteritis of ruminants and camelids caused by the bacterium *Mycobacterium avium* subsp. *paratuberculosis* (*Map*). In cattle, the disease typically causes intractable diarrhea and severe wasting in mature animals. Johne’s disease is a growing concern for dairy cattle farmers and the dairy industry, as the negative impact it has on the production and overall health of these animals is now well known [1]. At the same time, this disease also has a significant impact on beef cattle: in the most advanced stages, in fact, the sick subjects are strongly emaciated, with fluid diarrhea and can develop the so-called “bottle jaw”. These conditions can lead to the meat being declared non-compliant for human consumption following the post-mortem inspection by the official veterinarian, and in severe clinical stages, the animals are declared not fit for transport and therefore have to be killed on the farm [2,3]. In this context, various attempts have been made in Europe to develop programs for the control of the disease both at national and international levels [4]. To establish effective programs, it is of fundamental importance to obtain data regarding the prevalence of paratuberculosis in different geographical areas and in different types of farms. For example, higher prevalence areas show an animal-level serological prevalence up to 20%, while between-herd prevalence was found to be >50%; the average seroprevalence worldwide is reported from 3 to 5% [5,6]. Obtaining these data is not easy as the infection often tends to become chronic and there are no good methods for diagnosis in the early stages, which are often asymptomatic. Moreover, the farmer, after the initial zeal in adopting the given protocol, later abandons its application because it is deemed too demanding when compared to the apparently easier test-and-cull [7,8].

The so-called “management control programs”, “herd management practices”, “risk management plan”, and the “guidelines for the control” or the “manual for the control” focus on strategies that limit the exposure of calves to *Map* by avoiding both contact with adult cattle and their feces and, in more recent years, also by adopting aggressive culling practices [9,10,11,12]. Cow-calf producers seek to improve the future health and productivity of their herd by selectively culling animals that are diseased, producing sub-optimally and raising poor-growing offsprings. The culling policies of producers reflect, if only quantitatively, expectations of the future herd and individual animal health and performance. For JD, the biological basis of these expectations is influenced by the recurrence rate of disease conditions, such as diarrhea, production, and the probability of survival for another season. Research into cow-calf-culling decisions is centered on modeling optimal herd-culling policies. Unfortunately, the recommendations set in these programs are generally complex, the compliance of farmers is generally low, and it is, therefore, difficult to determine which recommendations are key elements for JD control and which have limited importance in this respect [1,13]. For instance, true-prevalence (TP) data extended to the expected countrywide/areawide cow-level or countrywide/areawide herd-level can be deceiving because of several critical issues intrinsic to the tests or to the sampling scheme [6]. These include disagreements from the same country or area, non-consistent sampling schemes, and incongruities between TPs calculated by authors and, most likely, test accuracy estimates [6]. The dairy cattle sector possesses a wide variety of structures and management practices, with smaller herds often using grazing for a larger part of the animals’ diet. The larger farms, on the other hand, often use a diet based more on silage and concentrates that are supplied to the animals directly in the barn. In this type of farm, calves are often raised separately from the adults or (in about half of the herds) are entrusted to independent structures.

The aim of this study was to investigate the effect of culling management practices on the seroprevalence of Johne’s disease in Holstein dairy cattle, in a similar environmental and managerial context.

## 2. Materials and Methods

### 2.1. Dairy Farms and Study Area

Fifty dairy herds in the Perugia and Assisi districts were visited. At the beginning of the study, the farms were randomly chosen from the national database (all the farm numbers were printed, cut out, divided into groups based on the area, and then randomly drawn). More numbers were chosen from areas with a higher density of farms in order to create a sample that was representative of the dairy industry of Umbria. The selected area extends approx. from the latitude 43.063611° to 42.92786° and the longitude from 12.200556° to 12.645067°. All farms were milking >20 cows and had participated in a study for the evaluation of paratuberculosis prevalence and a subsequent approach for the control of the disease during the previous 4 years. A commercial ELISA kit (IDEXX Paratuberculosis verification Ab test, IDEXX Westbrook, MN, USA) for the detection of serum antibodies against *M. avium* sub. *paratuberculosis* was used to test the samples. All samples were singularly tested. Then, samples with an S/P ratio between 0.15 and 0.30 were tested again using two wells. In the manufacturer’s description of the kit, it is stated that the “test shows a sensitivity in excess of 50% and specificity above 99%”. For the purpose of the screening, a herd had been considered positive with just one animal positive and enrolled in the subsequent study, which is the object of this work. Table 1 lists the 20 farms enrolled in the study according to inclusion criteria as shown in Table 2. On-farm audits were conducted and the management measures for calving barn, cow-calf separation, the rearing of calves and replacement heifers, the history of purchased heifers (if any), and bedding were assessed. Farms not complying with even one of these criteria were excluded. More in detail, all selected farms correctly identified each animal from birth, raised calves in areas of the dairy that were entirely separated from the milking herd, the time elapsed from birth to cow-calf separation was always less than 6 h, and the separation of areas for un-weaned calves, adult cattle, and effluent from adult cattle were adequate to prevent the paratuberculosis infection of younger animals. Heifer calves never grazed the same paddocks of adult cows and only at the 6th–7th months of gestation (at 20–22 months old) were they moved into the same paddocks. Each calf received the colostrum only from its dam for the first 3–5 days and then was fed a milk replacer with water available since birth. The supplement consisted of pellets and hay, the latter produced by the same farm. Weaning took place at the 10th week.

The audit form was completed at the same time as the sampling. Data were recorded using iOS devices with software HanDBase 4. After the audit period, all data were transferred to a computer. The files, saved as text (csv, comma-separated values), and then imported into FileMaker Pro v. 10 for Mac OS X, were used for data analysis.

### 2.2. Study Design and Sample Size

The dairy farms enrolled shared routine management practices but differed in the culling strategy adopted at the onset of the study, as shown in Figure 1. Adult animals were tested by ELISA and all positive were culled in both groups. All negative animals with early clinical symptoms (i.e., decreased milk production, progressive weight loss, and mild or periodic diarrhea [14,15]) were culled from the aggressive culling group (group A), while only those with positive fecal samples (culture and PCR) were culled from the non-aggressive culling group (group B). The average cow productive life, defined as the time from first calving to culling [14], was <1100 days in group A and greater (>1500 days) in group B.

The estimation of the minimum sample size of blood samples needed and the estimated prevalence of cases were calculated by applying two equations which were used and explained in detail by the same authors in a previous work [16].

Taking into consideration that most of the parameters in the equation were assumed and that the larger the sample size, the closer the possibility of a true estimation, more than 300 samples were analyzed per group.

### 2.3. Sample Collection

A total of 788 sera (366 from group A farms and 422 from group B farms) from 20 selected farms (3 from group A and 17 from group B) were used for this study. Samples were obtained from healthy adult animals older than 24 months of age. The volume of blood collected was about 10 mL. Blood was collected from the farm veterinarian during routine metabolic health-check procedures. A commercial ELISA kit (IDEXX Paratuberculosis verification Ab test, IDEXX Westbrook, MN, USA) for the detection of serum antibodies against *M. avium* sub. *paratuberculosis* was used to test the samples; sera were kept at refrigeration temperature and analyzed within 3 h from sampling. At first, all samples were singularly tested. Then, samples with an S/P ratio between 0.15 and 0.30 were tested again using two wells. In the manufacturer’s description of the kit, it is stated that the “test shows a sensitivity in excess of 50% and specificity above 99%”.

### 2.4. Data Analysis

True-prevalence (TP) was calculated using standard methods from apparent-prevalence (AP) using the sensitivity (se) and specificity (sp) claimed by manufacturer (se: 50%, sp: 99%) and lower values suggested by experience (se: 35%, sp: 95%) (Win Episcope 2.0, CLIVE, www.clive.ed.ac.uk, accessed on 21 March 2022). Farm geographic coordinates, when not available in the county dairy database, were calculated with the integrated GPS module on a Pentax K1-II digital camera (Ricoh Imaging Company, Ltd., Tokyo, Japan).

Logistic regression was used to compare the seroprevalence before and after the implementation of an aggressive culling strategy in farms from group A.

Multiple logistic regression was used to quantify the association of culling strategy and farm size with *Map* seropositivity with GraphPad Prism 8 for macOS (GraphPad Software, San Diego, CA, USA). The model is described by the formula: ln[P(Y = 1)] = Ln[(P(Y = 1)/P(Y = 0)] = β0 + β1 * B + β2 * C
where Y is the dependent variable (ELISA results); β0, β1, and β2 are the odds ratios for intercept, farm size, and type of culling; B and C are the independent variables (farm size and culling strategy); and P is the probability.

Variance Inflation Factors (VIF) were calculated from the final model to check for collinearity.

To analyze the risk, a stochastic simulation model was developed in Microsoft Office Excel 2019 with the use of the add-in «@Risk» v.8.1 for Excel (Palisade, Ithaca, NY, USA). @Risk is based on a Monte Carlo simulation that can provide beneficial outcomes and allow one to overcome uncertainty in quantitative analysis. A Monte Carlo simulation can perform risk analysis through the substitution of individual points of uncertain inputs with the distribution of possibilities. These are randomly tested over and over, for many iterations, and the model creates large sets of possible data which can then be further analyzed.

## 3. Results

The area object of this study has a population of about 8000 Holstein dairy cows distributed in approximately 90 farms, most of which are concentrated in the central part (Perugia and Assisi districts) where we find about 5000 subjects [17]. Farms in this area are typically newer free-stall. The herds rarely purchase replacement cows and heifers or bulls. Therefore, these herds have a low risk for bringing animals infected with Johne’s disease into the herds. Dairy farms in these regions also tend to cull from less than 15% to more than 35% annually.

Twenty dairy herds out of 50 audited were selected for this study, based on the consistency of managerial factors; inclusion criteria are shown in Table 2. The median herd size was 152 lactating cows in total (mean 211.2 range 20–410), 387 for farms with an aggressive culling strategy (group A, mean 307.7, range 126–410), and 29 for the others (group B, mean 53.47 range 20–166). The distribution of herd size after the Monte Carlo simulation is shown in Figure 2.

At first, logistic regression was conducted to establish whether the implementation of an aggressive culling strategy in group A resulted in a reduction in paratuberculosis seroprevalence after the 4-year test-and-cull approach. Table 3 indicates that the implementation of aggressive culling is not related to *Map* seroprevalence: OR 1.105 (95% CI = 0.612–1.997, *p* = 0.7408).

Based on ELISA results, for group A (*n* = 366 cows from 3 farms), 21 samples were positive (AP: 5.7%, TP: 9.7%, 95% CI 6.6–12.7%), while in group B (*n* = 422 cows from 17 farms), 14 samples were positive (AP: 3.3%, TP: 4.7%, 95% CI 2.7–6.8%) (Table 4). The logistic regression identified “herd size” as the factor apparently related to *Map* seroprevalence: OR 1.003 (95% CI = 1.001–1.005, *p* = 0.0169) (Table 5). The similarity in the odds ratios from the multiple logistic regression with the simple logistic regression indicates that there is little confounding effect of the “culling rate” on the relationship between the herd size and *Map* seroprevalence (Table 6). The OR from multiple logistic regression, however, being around 1, showed that both herd size and culling strategy had no effect on the probability of a cow being serologically positive.

Figure 3 shows predictions for a seroprevalence test to be positive based on the independent variables (herd size and culling strategy) and ignores the actual outcome. This graph shows the groups of negative and positive animals and the distribution of predicted probabilities for both of those groups. Looking at the violin plot for the group of negative animals, we can see that the majority of them had predicted probabilities of testing positive below 0.08 (with a median of 0.0435 and mean of 0.0440). When classifying the group of positive animals, the predicted probabilities are similarly uniformly distributed (with a median of 0.0644 and a mean of 0.0525).

Testing for multicollinearity showed that the two independent variables (herd size and culling rate) were not highly correlated with one another in the regression model given that the VIF was lower than 5 (Table 6, VIF = 3.570) [18].

The stochastic model gives no correlation between the expected *Map* seroprevalence and both herd size and culling type (Figure 4).

These results show that the blood samples of Holstein cows from farms with an average greater productive life were not more likely to test positive for *Map* antibodies than the blood serum of Holstein cows from farms characterized by a more aggressive culling strategy. Considering a test sensitivity and specificity of 50% and 99% respectively, as declared by the kit producers, the overall true serological prevalence in Umbria was calculated as 7.0% (95% confidence interval, 5.2% to 8.8%). Experience suggests that the sensitivity and specificity claimed by the manufacturer are optimistic. If sensitivity and specificity for the ELISA kit of 40% and 96% are assumed, then the true serological prevalence in Umbria dairy cattle overall is instead 1.2% (95% CI, 0.5%, 2.0%). With the positive cut-off point provided by the kit producers, and considering a herd positive when at least one animal was positive, 10 out of 20 herds (50%) can be considered positive. However, it is well-known that when a test with a specificity lower than 100% is used, the risk of falsely identifying an animal as positive is increased, and, by extension, the same risk is present at the herd level. Therefore, using a more rigorous standard might be advisable. For example, considering a herd positive when at least two animals are positive, 5 herds out of 20 (25%) could be classified as positive.

## 4. Discussion

The results proved that cows from farms with a longer cow’s average productive life did not have a statistically higher seroprevalence to *Map* antigens when compared to cows from farms with lower one due to the more aggressive culling strategy of animals showing early clinical symptoms. To control the spread of *Map*, test-based culling intervention is typically recommended [10,11,19]. Current diagnostic tests, such as fecal culture test, fecal PCR test, and ELISA have high test sensitivities for detecting animals shedding high levels of *Map*, but relatively low test sensitivities for detecting animals shedding low levels of *Map* [5,20].

The route of the transmission of JD is largely fecal–oral, with a very large number of organisms being shed in the feces by clinically affected animals. Lower bacterial counts are shed by infected animals without clinical signs of the disease [1]. It has long been recognized that the most susceptible animals are young calves and that it becomes increasingly difficult to infect cattle as they mature; notwithstanding, hygiene within the calving area, correct calving management, and early cow-calf separation measures are often difficult both to implement and to maintain over time. A common field approach for the control of *Map* infection is typically oriented toward the removal of already-infected animals and not to really prevent new infections among young animals, although they represent the most susceptible category [21]. Thus, the most-implemented practice on many dairy farms is to immediately cull animals showing early clinical symptoms, as they are considered to be a greater risk for spreading *Map*, as occurring in Group A farms in this study. Although some authors believe that the only control method that dairy farmers can apply to control paratuberculosis is culling and replacement [10,11,19], a test-and-cull strategy on its own, based only on the identification and elimination of clinically diseased and/or subclinically infected animals, is inadequate.

In fact, the available tests are insufficiently sensitive and most of all, the application of exclusively culling will simply fail to eradicate the disease [1,22,23]. For animals showing early clinical symptoms, culling is generally delayed or not done at all, as was likely to happen in group B in our study. Culling low-shedding animals may be more costly to the herd than the infections and losses they can cause [12,24,25], since replacement heifers are one of the largest costs in dairy operations, as also considered by Crociati et al. [26]. Group A and B farms did not differ for the seroprevalence of JD after 4 years of a test-and-cull policy (Group A) but differed in the average cow productive life, which strongly influences the overall farm net return and its long-term economic sustainability [27]. For these reasons, preventive management measures must be the starting point in a control program where a test-and-cull strategy is complementary to the whole management strategy. A first step in understanding both the decision process and consequences is to understand what the culling patterns are and what factors contribute to dairy cow culling [6,12,28,29].

Since non-management factors such as climate and soil type can influence the ability of *Map* to survive in the environment, it is obvious that risk factors also vary geographically [28]. The herein seroprevalence of Johne’s disease resulted within the range of apparent prevalence previously reported for Italy and other parts of the EU [6,30,31,32]. Aggressive culling practices (such as those used in northern Italy) might mitigate some economic effects [33], but until in-depth studies of the epidemiology and economics of Johne’s disease in dairy farms are performed, it remains to be determined whether *Map* has a large enough economic impact to justify a mandatory state-wide control program, rather than the current voluntary strategy that relies on individual producers [9]. In areas with high *Map* prevalence, it has been postulated to apply to all farms a program to prevent and control the disease [34].

To establish effective control programs, it is of fundamental importance to obtain data regarding the prevalence of paratuberculosis in different geographical areas and in different types of farms. Obtaining these data is not easy, as the infection often tends to become chronic and there are no good methods for diagnosis in the early stages, which are often asymptomatic. Reducing or avoiding the calves’ exposure to manure from positive adult animals is of great importance to minimize the risk of infection, along with the provision of clean feed and water for both young and adult animals. Another important point to avoid the diffusion of Johne’s disease is the management of colostrum and milk for the newborns; it is advisable to use only “low-risk” colostrum and milk, that is, colostrum and milk obtained only from healthy cows that were recently tested for paratuberculosis, or milk and colostrum directly tested for the presence of the pathogen [35]. This milk is usually kept separated from the rest to avoid the risk of contamination [17,36]. “Low-risk” colostrum and milk are often stored refrigerated or frozen in cattle farming. Another possible approach, useful especially if the herd presents high levels of positivity, is to use a colostrum or milk supplement. Another good practice, in order to minimize the risk of spreading paratuberculosis, is a thorough early separation of calves and cleaning of the udder and teats before milking [21,37,38].

## 5. Conclusions

This research represents an attempt to collect baseline prevalence data in Umbria, Italy for further considering the implementation of JD preventive measures in Holstein dairy cattle herds. This study suggested that an expensive test-and-cull strategy could be limited, but not erased, in farms that aim at reducing the seroprevalence of Johne’s disease. At the same time, correct calving and neonatal management should be re-prioritized and maintained over time.

## Figures and Tables

**Figure 1 vetsci-09-00162-f001:**
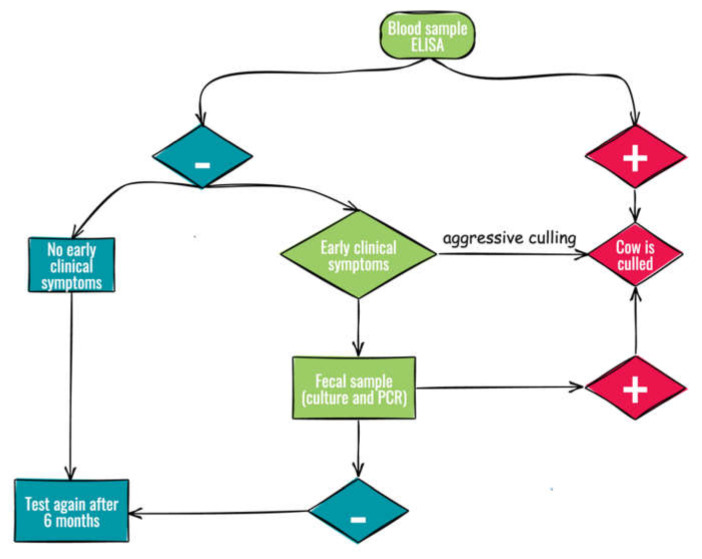
Flow chart of culling strategies for the control of Johne’s disease.

**Figure 2 vetsci-09-00162-f002:**
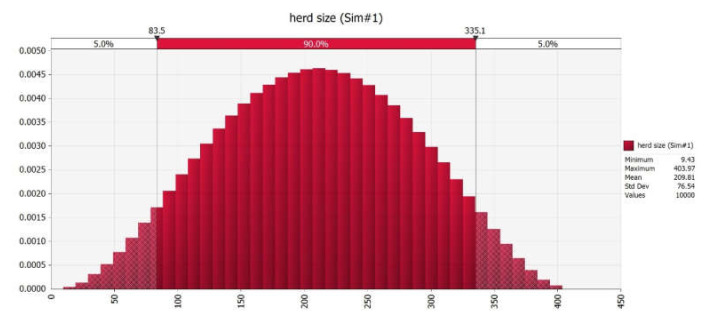
Distribution of herd size after Monte Carlo simulation.

**Figure 3 vetsci-09-00162-f003:**
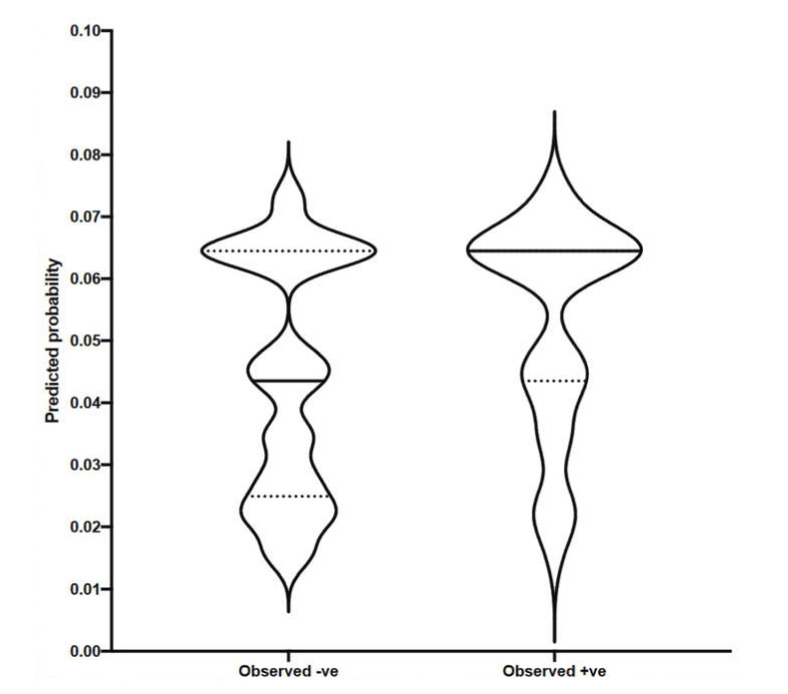
Multiple logistic regression, violin plot for predicted vs. observed.

**Figure 4 vetsci-09-00162-f004:**
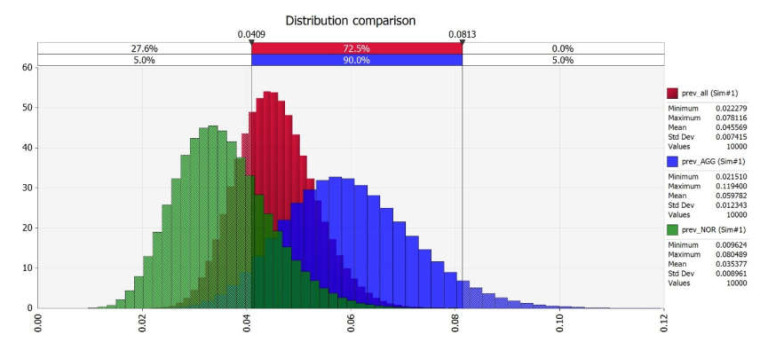
*Map* seroprevalence in all samples (red), in comparison with aggressive (blue) and normal (green) culling.

**Table 1 vetsci-09-00162-t001:** ELISA results in the 20 selected farms before the onset of the study, (S/P ratio of 0.30), AP: apparent prevalence; TP: true prevalence; CI: confidence interval.

Farm	*n*. of Adult Cows	*n*. of Cows Tested	% of Cows Tested	+ve	−ve	AP	TP	95% CI	
1	24	23	95.83	1	22	4.3	6.8	0.0	17.1
2	19	19	100.00	1	18	5.3	8.7	0.0	21.4
3	38	35	92.11	2	33	5.7	9.6	0.0	19.4
4	24	23	95.83	1	22	4.3	6.8	0.0	17.1
5	29	27	93.10	2	25	7.4	13.1	0.0	25.8
6	18	18	100.00	1	17	5.6	9.3	0.0	22.7
7	30	28	93.33	1	27	3.6	5.2	0.0	13.5
8	18	18	100.00	1	17	5.6	9.3	0.0	22.7
9	28	27	96.43	1	26	3.7	5.5	0.0	14.1
10	27	26	96.30	3	23	11.5	21.5	0.0	37.3
11	158	112	70.89	5	107	4.5	7.1	2.3	11.8
12	135	100	74.07	1	99	1.0	0.0	0.0	0.0
13	30	28	93.33	1	27	3.6	5.2	0.0	13.5
14	70	60	85.71	1	59	1.7	1.4	0.0	4.3
15	42	38	90.48	1	37	2.6	3.3	0.0	9.0
16	150	108	72.00	5	103	4.6	7.4	2.5	12.3
17	108	85	78.70	5	80	5.9	10.0	3.6	16.3
18	66	57	86.36	1	56	1.8	1.5	0.0	4.7
19	425	202	47.53	13	189	6.4	11.1	6.8	15.4
20	400	196	49.00	12	184	6.1	10.5	6.2	14.7
Total	1839	1230	66.88	59	1171	4.8	7.7	6.3	9.2

**Table 2 vetsci-09-00162-t002:** Responses to questions about management practices on 20 selected dairy farms in Umbria, Italy.

Audit Check List	Management Practices in the 20 Selected Farms
Area where calves are born	Dedicated maternity area
Time elapsed from birth to calf removal from its dam	<6 h
Adequate separation of area for unweaned calves from adult cattle and effluent from adult cattle	yes
Age when replacement heifers graze paddocks of adult cattle	20–22 months old (at 6–7 months of pregnancy, A.I. at 14–15 months)
Milk or milk replacer fed to calves	Colostrum for 3–5 days then milk replacer
frequency of feeding	bid
Method for feeding	Bucket with colostrum/milk from negative mothers or pathogen-free milk replacer
Bedding into the maternity area	Straw
Shelter	Shade—roof and sides
Source of water for unweaned calves	Town water, bore water, spring water
Water provided to calves from birth	Yes
Supplements	Yes
Type of supplements for unweaned	Pellets after 4–5 days and some hay
Age when calves receive the supplement	1–6 days
Source of hay	Both home (80%) and purchased (20%)
Age at weaning	<10 weeks
Source of water for weaned calves	Town water, bore water, spring water
Type of supplements for weaned calves	Pellets
Average productive life	Group A: <1100 days, Group B: >1500 days
Replacement heifers introduced	Yes (<5%), none during the study
Source of replacement	Private sales

**Table 3 vetsci-09-00162-t003:** Factors associated with *Map* seroprevalence in Holstein dairy cattle in Central Italy, results of a logistic regression before and after the implementation of an aggressive culling strategy in group A farms. OR: odds ratio.

Individual Variables	OR (95% CI)	*p*-Value
After implementation	1.105 (0.612–1.997)	0.7408

**Table 4 vetsci-09-00162-t004:** ELISA results in the 20 selected farms at the end of the study, (S/P ratio of 0.30), AP: apparent prevalence; TP: true prevalence; CI: confidence interval.

Farm	Average Productive Life	*n*. of Adult Cows	*n*. of Cows Tested	% of Cows Tested	+ve	−ve	AP	TP	95% CI	
1	Group B	21	20	95.24	1	19	5.0	8.2	0.0	20.2
2	Group B	22	10	45.45	0	10	0.0	0.0	0.0	0.0
3	Group B	42	18	42.86	1	17	5.6	9.3	0.0	22.7
4	Group B	25	14	56.00	0	14	0.0	0.0	0.0	0.0
5	Group B	27	12	44.44	1	11	8.3	15.0	0.0	35.2
6	Group B	20	8	40.00	0	8	0.0	0.0	0.0	0.0
7	Group B	27	11	40.74	0	11	0.0	0.0	0.0	0.0
8	Group B	20	4	20.00	0	4	0.0	0.0	0.0	0.0
9	Group B	25	3	12.00	0	3	0.0	0.0	0.0	0.0
10	Group B	29	11	37.93	1	10	9.1	16.5	0.0	38.5
11	Group B	166	73	43.98	3	70	4.1	6.3	0.8	11.9
12	Group A	126	61	48.41	1	60	1.6	1.3	0.0	4.2
13	Group B	33	12	36.36	0	12	0.0	0.0	0.0	0.0
14	Group B	75	35	46.67	0	35	0.0	0.0	0.0	0.0
15	Group B	49	22	44.90	0	22	0.0	0.0	0.0	0.0
16	Group B	152	73	48.03	4	69	5.5	9.1	2.5	15.8
17	Group B	110	69	62.73	3	66	4.3	6.8	0.9	12.8
18	Group B	66	27	40.91	0	27	0.0	0.0	0.0	0.0
19	Group A	410	45	10.98	3	42	6.7	11.6	2.2	20.9
20	Group A	387	260	67.18	17	243	6.5	11.3	7.5	15.2
	Total group A	923	366	39.65	21	345	5.7	9.7	6.6	12.7
	Total group B	909	422	46.42	14	408	3.3	4.7	2.7	6.8
	Total	1832	788	43.01	35	753	4.4	7.0	5.2	8.8

**Table 5 vetsci-09-00162-t005:** Factors associated with *Map* seroprevalence in Holstein dairy cattle in Central Italy: results of logistic regression for each variable. OR: odds ratio.

Individual Variables	OR (95% CI)	*p*-Value
Normal culling type	0.564 (0.282–1.125)	0.1041
Farm size	1.003 (1.001–1.005)	0.0169

**Table 6 vetsci-09-00162-t006:** Factors associated with *Map* seroprevalence in Holstein dairy cattle in Central Italy, results of multiple logistic regression. OR: odds ratio; VIF (variance inflation factor).

Individual Variables	OR (95% CI)	*p*-Value	VIF
Normal culling type	2.446 (0.412–14.525)	0.3249	3.57
Farm size	1.006 (0.999–1.012)	0.0734	3.57

## Data Availability

Data is contained within the article.

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
