# Peer review of "Effect of Culling Management Practices on the Seroprevalence of Johne’s Disease in Holstein Dairy Cattle in Central Italy"

_vetsci, 2022, doi:10.3390/vetsci9040162_

Round 1
Reviewer 1 Report
The authors addapted the manuscript according to the suggestions.
The numbering of table 3 and table 4 chould be exchanged, since table 4 is discussed in line 212 already whereas table 3 in 217.
I would prefer the inclusion of the information regarding the equations used to determine sample size in this manuscript to refering to another paper. However, if this leads to difficulties regarding plagiarism I can understand that the equations are not provided here. I will follow the editors decission in this.
Author Response
The numbering of table 3 and table 4 chould be exchanged, since table 4 is discussed in line 212 already whereas table 3 in 217.
AU: The tables’ numbering has been modified as suggested. The position in the manuscript was also modified.
I would prefer the inclusion of the information regarding the equations used to determine sample size in this manuscript to refering to another paper. However, if this leads to difficulties regarding plagiarism I can understand that the equations are not provided here. I will follow the editors decission in this.
AU: Thank you for your comment, we defer to editor for this decision.
Reviewer 2 Report
The authors have revised manuscript which addressed all comments and suggestions made by reviewers. The data analysis and interpretation are reviewed and I have no further comment.
Author Response
The authors have revised manuscript which addressed all comments and suggestions made by reviewers. The data analysis and interpretation are reviewed and I have no further comment.
AU: Thank you very much for your comment.
This manuscript is a resubmission of an earlier submission. The following is a list of the peer review reports and author responses from that submission.
Round 1
Reviewer 1 Report
Review
Effect of culling management practices on the seroprevalence 2 of Johne’s disease in Holstein dairy cattle in Central Italy
General comments:
The research question“ does culling strategy reduce seroprevalence in dairy herds“ is very interesting. However, the study approach and the data described here show considerable flaws.Sampling information and serologic findings per farm is missing. Defining a prevalence per area seems not convincing. The authors talk about a culling strategy for clinical cows. How is distiguished between culling because of johnes and culling due to other reasons? Of the 20 selected farms 3 belong to the group of aggressive culling and and 17 to the normal group. When cows are already clinical (which occurs only rarely in most herds) keeping them for milk production is highly unusual. Finding 17 out of 20 herds which keep clinical cows for some time in the herd is hard to believe. In addition, when group sizes differ so severely conclusions should be drawn very cautiously.
Point to point comments:
Line 43-45: This condition is also true for dairy cattle.
Line 45-47: Cattle in that stage of the disease is no longer suitable for human consumption. Sending it to slaughter anyway is besides a consumer risk also an animal welfare issue. Please rephrase that sentence.
Line 52-55: repetition of „while“ please rephrase.
Line 82-85: What do you mean with this sentence? If you have a closed farm and use appropiate measures they are not effective because the neighbour is making a mess out of it? I cannot believe that. We are talking about bacterium here and not a virus.
Line 96-101: This information does not belong tot he introduction. Result section?
Line 102-105: Please explain your objectives more clearly. How can the information regarding the culling strategie of a farmer be of any help to re-prioritize the effort to improve the correct calving area and calving management practices?
Line 105-108: Did you study the difference of seroprevalence in realtion to a longer productive life or in relation to a certain culling strategie oft he farmer? These are two different qbjectives.
Line 112-118: How many farms are located in that area. Was the size included in the randomization process? Do all farms are known tob e infected with MAP? Do you have information regarding their MAP-prevalence?
Line 124-125: What does „inclusion criteria were represented by the most diffuse management practices observed among the 50 dairy herds“ Please explain.
Line 119-125: So from the 50 randomly selected farms you selected 20 according to your selection criteria?
Line 127-135 should be moved to the result section.
Line 141-145: Ist he aggressive culling policy only used for cows suspicious for MAP infection? It seems to me that there are a lot more reasons on a dairy farm to cull a cow which occur more often than a clinical symptoms of johnes disease. How did you differentiate between culling reasons?
Line 149-153: Please provide the information regarding the equations in this manuscript as well. Sample size needed for what? So did you calculate how many samples you needed or did you take and analyze more than 300 (so how many exactly) samples per group just to make sure?
Line 156-157: How many samples per farm were selected? How did you select the cows which were sampled? Three farms of group A and 17 of group B makes a comparison of the results difficult.
Line 166-168: Did you calculate the TP per farm or for the area?
Line 169-172: You devided the farms in Group A and B for your analyses and now you add up all your results to calculate a TP for an area and extrapolate the result to all farms oft he area. Seems not correct to me.
Line 172-179: DId you analyse the association of culling for all 13 farms in that area? How do you know which culling strategy is addopted on which farm?
Figure 3: please provide explanations for figure 3. What represents the X and the Y axis? It seems that the mean prevalence is lower in the green herds compared tot he blue herds?
Line 271-273: What does „diffuse practice“ mean?
Line 300-317: This paragraph hs not been part oft he research. The conclusion to re-prioritize efforts form culling policy may be correct, owever, I cannot see how the results of this study lead to these recommendations.
References:
- Rosemberger: The author is called „Rosenberger“. However, please refer to scientific publications in your manuscript and so not cite textbooks which are over 40 years old.
Reviewer 2 Report
Authors set a very clear research question in a very complicate problem like the paratuberculosis prevalence and the measures for eradication or reduction of it in dairy cows’ farms.
It is an interesting paper, with clear goals and useful approach for clinical practice. Interesting result that culling rate does not influence the positive paratuberculosis animal is an important finding and it could be stresses even more in the text.
Introduction is by far too extended (over 100 lines), for such a well known problem as the paratuberculosis in dairy farms. It is suggested to be shortened to almost half in extend, as to be more accurate and stress better the study design and aims.
In the same logic, conclusion part is also too long. Conclusion could be limited in L325-328, which is adequate for conclusions with the addition of the calculated paratuberculosis prevalence in Umbria, which was one of the study aims.
The most important comment is about the sampling procedure. Authors claim that harvested blood sera from 788 animals from the selected farms (L155-2.3. Sample collection). It should be clearly stressed the age of the sampled animals. For paratuberculosis seroprevalence investigation only for animals older than 24 months (or 30-36 months, depending on epidemiological schedule) are sampled. In case authors didn’t followed this and sampled random animals without age criterion, then the whole concept is wrong and the presented paper cannot be accepted. In table 2 is mentioned the number of adult cows and the number of cows sampled. It can be assumed, which is what I also understand, that authors sampled only adult cows. It is important that this should be clearly added in the text and especially in the 2.3 sample collection part. Authors should stress that they samples only adult cows and define that age e.g. over 24 or 28 month of age, of the sampled animals. It is also important to provide (if available) information about the presence or absence of clinical manifestation of paratuberculosis in sampled animals.
Other comments:
Citation is not necessary in abstract (L29). It suggested to be omitted and adapt accordingly the other citation numbers in text.
L64 needs correction in language
L88-90 needs correction in language
212-213 language improvement as to be comprehensive
L288 typo-mistake
Reviewer 3 Report
Practical difficulties to implement and sustain good hygienic practice and herd health management to prevent JD infection has been highlighted. Challenges of determining culling practices for JD control have been elaborated.
The study design is meticulously planned and executed. Sample size estimation and culling criteria have been described. Deficiency of current diagnostic tests and conventional test and slaughter policy has been pointed out. Good hygienic management of feeding colostrum and milk to newborn has been highlighted to prevent JD transmission in farms. The authors have highlighted difficulty in data collection to determine the prevalence of paratuberculosis in different geographical areas and in different types of farms and lack of highly sensitive diagnostic tests in early stage of JD infection which needs to be addressed in future.
There are few queries to be addressed by authors;
The study on sero-prevalence of Johne's disease (Map) expected that whether Holstein dairy cows from the farm with longer productive life (lower culling rate with productive life greater than 1500 days) ( study group B) were more likely to be serologically positive to Map rather than those cows from the farm with lower productive life (less than 1100 days) due to aggressive culling strategy (study group A) in a similar environmental and managerial context. The query is that whether we can say these two groups of animals under study have similar managerial context! Would be better to clarify (because the productive life in both groups is different).
Group A constituted the farms where aggressive culling policy was adopted. This was the group (A) in which sero-positive were found to be higher (in the present study) than those of Group B farms where lesser aggressive culling policy adopted. Here the query is that why the Group A farms adopted aggressive culling policy? Did these farms (Group A) already had higher sero-prevalence rate of Map as compared to the Group B animals? It may be possible that higher prevalence of Map could force Group A farms to adopt aggressive culling policy??? Are there any past data on the sero-prevalence of Map for both Group A and Group B farms available? Would be better if simply justify with logical statements of these queries.